# Motility phenotype in a zebrafish *vmat2* mutant

Hildur Sóley Sveinsdóttir[1], Amanda Decker[2], Christian Christensen[1], Pablo Botella Lucena[3¤], Haraldur Þorsteinsson[1], Elena Richert[3,4], Valerie Helene Maier[5], Robert Cornell[2], Karl Ægir Karlsson [1,3,5]*

**1** 3Z, Reykjavik, Iceland, **2** Department of Anatomy and Cell Biology, University of Iowa, Iowa City, Iowa, United States of America, **3** School of Science and Engineering, Reykjavik University, Reykjavik, Iceland, **4** Department of Psychology, University of Oldenburg, Oldenburg, Germany, **5** Biomedical Center, University of Iceland, Reykjavik, Iceland

¤ Current address: Department of Neurosciences, Biomedical Research Institute, Hasselt University, Hasselt, Belgium

* karlsson@ru.is

**Data Availability Statement:** All relevant data are within the paper and its files.

**Funding:** This work is funded in part by grants from the National Institutes of Health GM067841 (RAC) and AR062547 (RAC). NO The funders had

## Abstract

In the present study, we characterize a novel zebrafish mutant of *solute carrier 18A2* (*slc18a2*), also known as *vesicular monoamine transporter 2* (*vmat2*), that exhibits a behavioural phenotype partially consistent with human Parkinson´s disease. At six days-post-fertilization, behaviour was analysed and demonstrated that *vmat2* homozygous mutant larvae, relative to wild types, show changes in motility in a photomotor assay, altered sleep parameters, and reduced dopamine cell number. Following an abrupt lights-off stimulus mutant larvae initiate larger movements but subsequently inhibit them to a lesser extent in comparison to wild-type larvae. Conversely, during a lights-on period, the mutant larvae are hypomotile. Thigmotaxis, a preference to avoid the centre of a behavioural arena, was increased in homozygotes over heterozygotes and wild types, as was daytime sleep ratio. Furthermore, incubating mutant larvae in pramipexole or L-Dopa partially rescued the motor phenotypes, as did injecting glial cell-derived neurotrophic factor (GDNF) into their brains. This novel *vmat2* model represents a tool for high throughput pharmaceutical screens for novel therapeutics, in particular those that increase monoamine transport, and for studies of the function of monoamine transporters.

## Introduction

The Vesicular monoamine transporters are membrane-embedded proteins that transport monoamines into synaptic vesicles using a proton gradient [1]. The two paralogs (i.e., VMAT1 and VMAT2, renamed to Solute Carrier 18 A1 and A2 (SLC18A1 and SCL18A2)) are differentially expressed, where VMAT1 is mainly expressed outside of the central nervous system in endocrine cells, chromaffin and enterochromaffin cells [2] and transports serotonin, epinephrine and norepinephrine [3]. VMAT2 on the other hand is primarily expressed in the central nervous system [4] and transports all monoamines into synaptic vesicles [5]. VMAT2 therefore

no role in study design, data collection and analysis, decision to publish, or preparation of the manuscript.

**Competing interests:** I have read the journal's policy and the authors of this manuscript have the following competing interests: KÆK and HÞ are co-founders and shareholders in 3Z. This does not alter our adherence to PLOS ONE policies on sharing data and materials.

reduces cytosolic dopamine, which is important in normal cellular function and prevents neuronal damage through accumulation of dopamine outside vesicles [6]. Changes to VMAT2, is associated with various disorders including depression, drug addiction, anxiety, stress, and Parkinson disease (PD) [6–9]. Significantly reduced VMAT2 mRNA expression for example has been seen in brains of humans with PD [10]. Further, the loss of VMAT2 was found to be a key pathogenic event in a non-human primate model of PD [11] and, in congruence, reduced *VMAT2* expression lead to a stronger phenotype in rodents in a neurotoxin induced PD model [12] whereas *VMAT2* overexpression was protective [13,14].

*VMAT2* expression has been modified in animal models to pursue a better understanding of the motor and non-motor dysfunctions related to PD. Previous studies in mice have demonstrated that complete loss of *Vmat2* is fatal, while mice with a 95% reduction in *Vmat2* expression are viable and display dopamine depletion, loss of DA neurons and the build-up of α-synuclein [15,16]. Jointly, these studies have documented both motor and non-motor symptoms of PD in various assays including assays for strength, gait, olfaction, and sleep [15,16]. The peptide sequences of zebrafish Slc18a2 and human SLC18A2 are 75% identical [17]. Expression of zebrafish *vmat2* overlaps with expression of *tyrosine hydroxylase 1* (TH), a molecular marker for DA, throughout the larval zebrafish brain [18]. The ventral diencephalon corresponds functionally to the *substantia nigra* and ventral tegmental area in mammals [19–24], the key area of dopamine-loss coinciding with PD [25]. Expression of a truncated form of *vmat2* in zebrafish, generated using CRISPR-Cas9, revealed a phenotype of juvenile zebrafish with increased anxiety-like behaviour in novel tank assessment and dark avoidance in light-dark box test compared to controls [26].

A novel zebrafish model of PD is thus poised to be valuable for the understanding of the disease in general and in the quest for novel PD therapeutics in particular. Recently a novel zebrafish *vmat2* mutant model has been generated using CRISPR/Cas9 and characterised in detail [27]. Zebrafish carrying a knockout mutation for vmat2 were found to exhibit low levels of dopamine, noradrenaline, 5-hydroxytryptamine and histamine; the biogenic amines transported by Vmat2 and, consistently, lower number of dopamine, 5-hydroxotryptamine and histamine immunoreactive neurons. At the same time amine-synthesising genes were upregulated and the mutant vmat2 fish had higher levels of dopamine metabolites, suggesting increased monoamine turnover in the mutants [27].

In the present study our first aim was to characterize the behavioural phenotype of these *vmat2* mutant zebrafish larvae. Our second aim was to assess if the *vmat2* mutant larvae are a suitable model for drug discovery of novel PD therapeutics by administering selected dopamine precursor and agonists (L-Dopa and pramipexole, which also has a low affinity to serotonin [28]), as well as glial cell line-derived neural growth factor (GDNF), a putative neurorestorative agent [29].

## Materials and methods

### Fish and generation of mutant

In this study a zebrafish (*Danio rerio*) *vmat2*$^{ui105}$ mutant strain was used, generated in an outbred background at the University of Iowa as described previously in [27]. This mutant line carries a 5-basepair deletion in exon 3 of *vmat2* leading to a frameshift and predicted knockout of the corresponding protein. Homozygous mutant fish (*vmat2*$^{-/-}$) were detected in the larvae, but not viable as adults, therefore heterozygous *vmat2* (*vmat2*$^{-/+}$) fish were kept as a breeding group. The general physical condition of the heterozygous *vmat2* mutant fish was not affected by the mutation and they looked, ate and aged as wildtype fish in the facility. For behavioural analysis a total number of 1826, 6 days-post-fertilization (dpf) larvae were used in this study.

Zebrafish were maintained at Reykjavik University and fed three times a day on a variable diet of TetraMin flakes (Tetra Holding GmbH, Melle, Germany), Adult Zebrafish Complete Diet (Zeigler Bros, USA) and live Artemia (INVE Aquaculture Nutrition, USA). Fish were kept in a 14:10 light:dark cycle in 3 or 10 L multi tank constant flow system (Aquatic Habitats, Apopka, FL, USA). Water temperature was held at a constant 28.5˚C and replaced at a rate of 10% per day. Eggs were collected between 9:00–11:00 hrs and contained in 2 L tanks. The following day, dead eggs were removed and tanks were cleaned. Eggs were incubated for 4 days at 28.5˚C in system water mixed with methylene blue (2 ml 0.1% methylene blue per 1 L). All procedures in this study were carried out in strict compliance with the regulations of, and approved by, the National Bioethics Committee of Iceland (regulation 460/2017).

## Behavioural recordings

At 5 dpf, larvae were placed in individual wells of 96-microwell plates or 24-microwell plates (Nunc, Roskilde, Denmark) in system water. The microwell plates were relocated to a custom-built activity monitoring system fitted with 24 infrared cameras (Ikegami, ICD-49E; Ikegami Tsushinki Co, Japan) which was thermo-regulated at 28.5˚C, blocked from daylight and illuminated from below with white (255 lx; light-phase) and infrared light (0 lx; dark-phase). Larvae, in all assays, were left to acclimate in the activity monitoring system for 24 h prior to recording. At 6 dpf, behaviour was tracked in two dimensions at 5 Hz. An exclusion criterion was based on the percentage of samples during a recording where a larva was not tracked. The threshold was set to 10%, thus a larva that was tracked <90% of the total recording time was excluded.

## Motor assay

Recordings, performed in 96-well plates, started at 13:00 with the lights on and continued with alternating dark and light phases in 30-minute intervals between 13:30 and 18:00. Two parameters were examined during a 60-minute phase from 13:30–14:30, which included 30-minute dark phase, 30-minute light phase and the transition from light-to-dark. First, peak velocity (mm/s) was defined as the velocity at the time of first transition from light-to-dark, that is, the average velocity measured for a 30 second period immediately following the change from light to dark. Second, distance moved (mm) was defined as the distance moved during the first 30-minute dark phase followed by 30-minute light phase.

## Sleep assay

Sleep behaviour was recorded in a 96-well plate and analysed during the lights-off period (22:00–08:00) and during a 30-min segment following lights-on (08:05–08:35). Sleep in freely moving zebrafish is defined solely using behaviour [30]. First, all behaviour was dichotomized into 1-s bins of movement or non-movement. Prior, in-depth frame-by-frame video analysis by three independent evaluators resulted in the adoption of the speed of 1.0 mm/s as the threshold for movement for larval zebrafish. All activity that was slower than that threshold was computed as non-movement. Thus, rendering a dichotomized record of the behaviour in either movement or non-movement. Next, the dichotomized record was transformed into bins of sleep and wake. Following previously established sleep criteria in adults, and adapted to larval fish [31–34], six or more consecutive 1-s bins of non-movement were counted as sleep and all else was counted as wake. That is, the seventh second and above were classified as sleep; all other bouts were classified as wake. Once the number of sleep and wake bouts was calculated, five different sleep parameters were assessed. Sleep fragmentation was defined as the number of transitions between sleep and wake bouts per hour. Sleep ratio was calculated as the

percentage of total night time that the fish was considered asleep. Sleep velocity (mm/s) was defined as the average velocity throughout the night time. Wake bout duration (s) was defined as the average length of wake bouts. Sleep bout duration (s) was defined as the average length of sleep bouts.

## Thigmotaxis assay

Thigmotaxis, indicative of anxiety, is a behaviour characterised by avoidance of an arena centre and a preference for the close proximity of boundaries in a novel environment [35,36]. This has been demonstrated across species, and in zebrafish, thigmotaxis can appear in larvae as young as 5 dpf, where the larvae prefer to move in contact with or near the edge of the arena, such as the wall of a well plate [37]. Thigmotaxis was measured in 24-well plates between 18:00 and 22:00 during lights-on and defined as the larva´s distance from centre point (mm) of the well to the wall.

## Genotyping

In this study heterozygous *vmat2* mutant zebrafish were used for breeding, therefore the offspring were genotyped and classed into wild type, heterozygous and homozygous. After behavioural recordings, the water was removed from the wells and plates were frozen at -80˚C until genotyped. Genomic DNA was extracted using Lysis Buffer (10 mM Tris pH 8, 1 mM EDTA, 0.01% SDS, 100 mM NaCl, 100 µg/ml Proteinase K) and incubation at 55 C for 3 hours. KASP genotyping was performed according to the manufacturer's recommendation (LGC, UK). The following sequence containing the site of interest in exon 3 of *vmat2* was supplied to the company. CTAGTGCCTATTATCCCAAGTTACCTGTACACGGTGGACGACGAGGCTGC[TCAGA]TGGTTAAGAATCACTCCATGACCCCTCTTTCTCCATCGAGCACCTTTCAG. Primers for KASP by design (KBD) were custom designed against this region and contained two different dyes (FAM-dye or HEX-dye) to detect the difference between wild type, heterozygous or homozygous mutant larvae. In short, zebrafish DNA was diluted to 5-10ng/µl and mixed with the KASP master mix (LGC, UK) and primers as recommended by the manufacturer. Samples were run in 96-well plates in a DNA engine tetrad PCR machine (BioRad). The cycling program started with 15 min at 94˚C, and subsequently 10 cycles of 20 seconds, denaturation at 94˚C and 1 min annealing/extension at 61˚C. A second set of 28 cycles was performed with 20 seconds at 94˚C and 1 min annealing/extension at 55˚C. Plates were read in an Applied Biosystem 7500 Real Time PCR machine. PCR and samples were grouped into the three genetic groups wild type, heterozygous and homozygous using the SDS 2.0 software.

## Imaging

In order to determine the number of TH positive cells in *vmat2* mutant larvae, 5 dpf larvae were cut behind the pectoral fins and heads were used for imaging, while the tail of the respective larvae was genotyped as described above. Staining was performed using the nuclear stain DAPI (Sigma-Aldrich, St. Louis, USA) and the Anti-Tyrosine Hydroxylase antibody (ab112, abcam, UK) labelling dopamine cells. This antibody is thought to recognise both TH1 and TH2, but several studies have suggested that the tyrosine hydroxylase detected in zebrafish brain is predominantly TH1 [38,39]. Sixty-eight heads of larvae were fixed in 10% buffered formaldehyde and subsequently permeabilised in ice-cold acetone. Samples were blocked in PBDT buffer (1% DMSO, 1% BSA, 0.5% TritonX-100, 1xPBS) containing 5% normal goat serum (NGS) for 1 h and then incubated with -TH antibody (1:500 in 5% NGS/PBDT) overnight at 4˚C. After washing, larvae were incubated with both DAPI (1:1000 in 2% NGS/PBDT) and 488 Alexa conjugated goat anti-rabbit antibody (1:500 in 2% NGS/PBDT). After a final

wash in PBS, larvae were mounted using Gold-antifade mounting liquid (Thermo Fisher Scientific Inc.). Fluorescent images were acquired using an Olympus IX33 confocal microscope (Olympus, Tokyo, Japan). The emission ranges were selected at 505–525 nm. The whole brain was visualized for cell counting using a 30X objective. Image stacks were individually analysed and cells of the diencephalic populations 5,6 and 11 [19–24] counted using (multipoint counter) ImageJ Fiji software [40]. Cell counting was performed blinded to the sample group.

## Drug preparations

The larvae were exposed to two different drugs during the behavioural recordings, pramipexole dihydrochloride and L-Dopa ethyl ester (both Sigma-Aldrich, St. Louis, USA, CAS: 104632-25-9 and 37178-37-3). Drug preparation was performed at the day of recording. Drugs were diluted using distilled water to reach their respective concentrations. For pramipexole and L-Dopa, larvae were submerged in 1 μM, 10 μM and 100 μM doses. Drugs were administered into the wells between 11:30 and 12:30 on the day of recording.

## GDNF

Effects of GDNF were assessed using human recombinant GDNF (Icosagen, Össu, Estonia) at a concentration of 1 ng/nl. GDNF was administered as brain injections using a Drummond capillary (inner diameter 0.53 mm x outer diameter 1.14 mm) and Drummond Nanoinjector II (Drummond Scientific, Brookmall, USA) at 4 dpf. Capillaries were heated and pulled for a sharp tip (Narishige PC-10, Narishige, Japan) and inserted midline, post-ocular and aimed at the tectal ventricle. Preliminary injections using dye were performed to verify the anatomical location and repeatability of injections. A total volume of 9.2 nl was injected, corresponding to 9.2 ng GDNF per larva. To rule out effects due to injection of liquid, sham injections with distilled water of the same volume were performed. Injections took place between 10:00 and 13:00 for GDNF and 14:00 and 17:00 for distilled water. Prior to injection, larvae were sedated in 0.01% MS-222 (Sigma-Aldrich, St. Louis, USA). After injection, larvae recovered in system water for approximately 48 h. At 6 dpf larvae were placed in the recording system described above. After the 24 h acclimation period, data recording was performed on 7 dpf for 20 h.

## Data analysis

Data was obtained using EthoVision XT (Version 11.5.2016, Noldus) and exported into Microsoft Excel for analysis. Sleep parameters were calculated using a custom written software. Statistical analyses were performed using IBM SPSS Statistics for Windows, version 26 (IBM corp., Armonk, N.Y., USA) and GraphPad Prism Software (Version 9.0.0, GraphPad Software Inc.). Figures were produced using Microsoft Excel and GraphPad Prism Software. Statistical differences between genotypes for: peak velocity, distance moved, thigmotaxis, number of TH positive cell and sleep were evaluated using one-way ANOVA and Bonferroni post hoc analysis. Data for behavioural analyses, including peak velocity, distance moved and sleep, were pooled from five independent experiments. Data for thigmotaxis were pooled from two independent experiments. To determine statistical differences between genotypes treated with pramipexole and L-Dopa, data on peak velocity, distance moved and sleep were evaluated using one-way ANOVA and Dunnett's two-sided post hoc analysis. For analysis of the data from larvae undergoing GDNF injections and sham injections, pooled from three independent experiments, statistical differences were evaluated using two-way ANOVA and Tukey´s post hoc analysis. $P < 0.05$ was considered statistically significant. All data are presented as mean ± standard error (s.e.m.).

## Results

### Homozygous *vmat2* mutant larvae express a multifaceted motility phenotype

Zebrafish larvae respond to abrupt lights-off with a burst of motility followed by a period of reduced movement. Overall, motility in all larvae during the 10-hour lights-off phase was reduced compared to the lights-on phase (Fig 1A). When comparing homozygous mutant larvae (*vmat2*$^{-/-}$), heterozygous mutant larva (*vmat2*$^{-/+}$) and wild-type (WT) sibling controls, homozygous mutant larvae exhibited a stronger motility response to the transition from lights-on to lights-off than did heterozygous or wild type larvae, but lower overall motility during the subsequent lights-on phase as depicted in an expanded view of the first light transition cycle (Fig 1B). One-way ANOVA revealed a statistically significant difference between genotypes for peak velocity ($F(2,166) = 49.007$, $p < 0.001$) during the transition from lights-on to lights-off. We found that homozygous mutant larvae showed significantly higher peak velocity levels compared to both heterozygous (2.29 ±0.11 mm/s vs 1.25 ± 0.07 mm/s, $p < 0.001$) and wild type (2.29 ±0.11 mm/s vs 1.22 ± 0.07 mm/s, $p < 0.001$) larvae (Fig 1C). Comparison of the average distance moved revealed a significant difference between genotypes during the 30 min lights-on ($F(2,166) = 14.675$, $p < 0.001$), but not during 30 min lights-off ($F(2,166) = 0.644$, $p = 0.526$). Homozygous mutant larvae moved significantly less than both heterozygous (730.5 ± 43.7 mm vs 1070 ± 44.3 mm, $p < 0.001$) and wild type larvae (730.5 ± 43.7 mm vs 922.2 ± 50 mm, $p = 0.029$) during lights-on (Fig 1D). These results suggest that homozygous mutant larvae express both a hypo- and hyper-motile phenotype depending on the behavioural parameter measured.

### *Vmat2* mutant larvae display an anxiety-like phenotype

To determine the effect of the genotype on anxiety-like behaviour of zebrafish larvae, we assessed thigmotaxis, defined as a preference for dwelling in the periphery over the centre of the well. Comparing the distance of the larvae from the centre point revealed significant differences between genotypes ($F(2,169) = 15.08$, $p < 0.0001$). We found that homozygous mutant larvae tended to dwell closer to the edge of the well than heterozygous (6.64 ± 0.09 mm vs 5.99 ± 0.08 mm, $p < 0.0001$) and wild type (6.64 ± 0.09 mm vs 5.96 ± 0.08 mm, $p < 0.0001$) larvae (Fig 1E). These findings indicate higher anxiety-like behaviour in larvae with a presumptive loss of Vmat2 protein. An explanation for the behavioural phenotypes seen in our study may be that loss of Vmat2 protein reduces vesicular sequestration of monoamines, resulting in less dopamine (or other biogenic amines) being dispersed in the synaptic cleft and a potentially cytotoxic accumulation of it the cytoplasm.

### *Vmat2* mutation results in day-time hypersomnia, but does not alter sleep parameters during night

First, sleep parameters were examined during the 10 h lights-off phase and contrasted between all three genotypes. There were no significant differences between genotypes in any of the sleep parameters examined (i.e., fragmentation, ratio, velocity, wake bout duration or sleep bout duration) (Fig 1F). Second, sleep parameters were examined during a 30-min segment following lights-on (after the night recording) and contrasted between all three genotypes. Comparison of the sleep ratio revealed a significant difference between genotypes ($F(2,166) = 3.654$, $p = 0.028$), where homozygous *vmat2* mutant larvae exhibited a significantly higher sleep ratio than wild types (0.407 ± 0.017 vs 0.332 ± 0.025, $p = 0.049$) (Fig 1G), indicating a day-time hypersomnia in mutants.

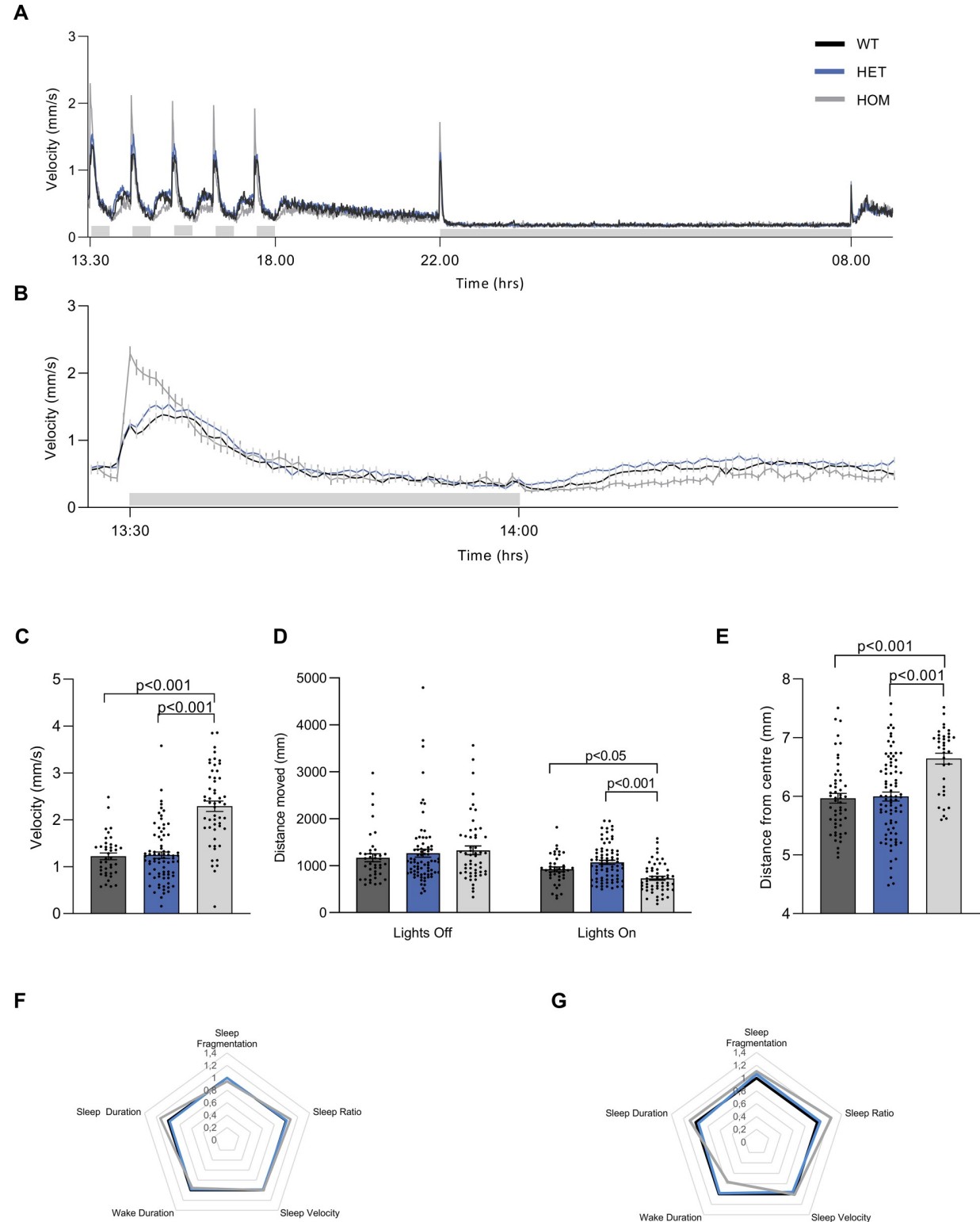

**Fig 1. Overview of behavioural parameters of *vmat2* larvae. (A)** Swim velocity of homozygous wild type (WT, n = 40), heterozygous *vmat2* (HET, n = 76) and homozygous *vmat2* (HOM, n = 53) larvae depicting the 30-minute intervals of lights-off (grey shaded bar) and lights-on phases between 13:30 and 18:00, followed by constant light-on from 18:00 to 22:00. Lights were turned off for the night (grey shaded bar) from 22:00 to 8:00 the next morning when the lights were turned on again. **(B)** Swim velocity of homozygous wild type (n = 40), heterozygous *vmat2* (n = 76) and homozygous *vmat2* (n = 53) larvae during the first hour of lights-off (grey shaded bar) and lights-on. **(C)** At the onset of the first

lights-off phase, peak velocity was analysed and it was revealed that homozygous *vmat2* larvae exhibited a higher peak velocity than wild type and heterozygous larvae. **(D)** The total distance moved during the first 30-minutes of lights-off and the following 30-minutes of lights-on demonstrates that homozygous *vmat2* larvae moved significantly less during lights-on than wild type or heterozygous larvae, while there was no difference during lights-off between all three strains. **(E)** Thigmotaxis was evaluated for wild type (n = 53), heterozygous *vmat2* (n = 83) and homozygous *vmat2* (n = 36) larvae as the distance of the larvae from the well centre during lights-on between 18:00 and 22:00. Homozygous *vmat2* larvae tended to dwell closer to the edge of the well than heterozygous and wild type larvae. **(F)** Sleep parameters in wild type, heterozygous *vmat2* and homozygous *vmat2* larvae were examined during the night, 10-hour lights-off phase. No significant differences between genotypes were observed in any of the sleep parameters. **(G)** Sleep parameters in wild type, heterozygous *vmat2* and homozygous *vmat2* larvae were examined during a 30-minute lights-on segment following the 10-hour lights-off, homozygous *vmat2* larvae exhibited significantly higher sleep ratio during this time. Sleep parameter data are represented as fold change of wild type larvae.

### *Vmat2* mutant larvae have reduced number of TH positive cells

DA cells were identified by immunostaining for TH and positive cells were counted in the midline diencephalic catecholamine monoaminergic cluster (groups 5,6 and 11 [20–24,41]) (Fig 2A and 2B). These results showed significant difference between genotypes (F (2, 26) = 13.93, p<0.0001). We found that DA cell counts were significantly lower in homozygous *vmat2* mutant larvae as opposed to wild type (18.44 ± 2.10 cells vs 31.17 ± 1.86 cells, *p*<0.001) and heterozygous (18.44 ± 2.10 cells vs 30.88 ± 1.61, *p*<0.001). No difference was found between wild type and heterozygous larvae (Fig 2B). In all the preceding assays (motor, sleep, thigmotaxis and cell count) no differences were found between heterozygous and wild type larvae, therefore, heterozygous larvae were excluded from further analysis. A partial explanation for the behavioural phenotype observed here might therefore be a reduction of DA cells in homozygous *vmat2* mutant larvae.

### The effects of dopamine agonist and dopamine precursor on behavioural phenotype in *vmat2* mutant larvae are not consistent with a PD phenotype

To gauge if the phenotype is indeed driven by dopamine cell reduction, we turned to two medications used to treat Parkinson's, the dopamine precursor levodopa (L-Dopa) and the dopamine agonist pramipexole. When examining motor parameters neither pramipexole nor L-Dopa significantly altered the peak velocity (Fig 3A and 3B) of larvae of the different *vmat2* genotypes (Table 1). Interestingly, pramipexole augmented the reduction between wild type

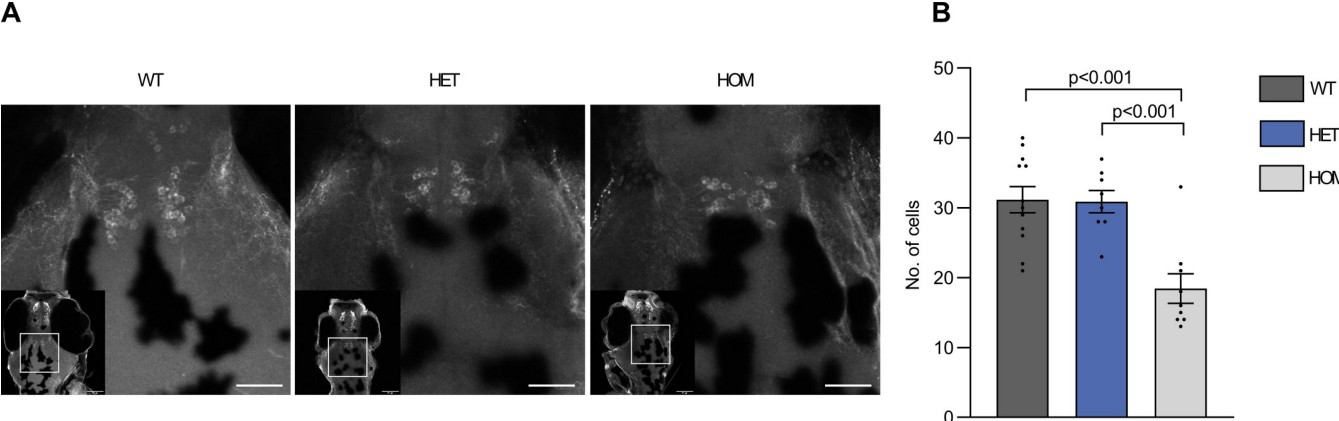

**Fig 2. Numbers of dopamine cells in *vmat2* larvae. (A)** The diencephalic populations 5,6 and 11 were examined after staining of homozygous wild type (WT), heterozygous *vmat2* (HET) and homozygous *vmat2* (HOM) larvae for Tyrosine Hydroxylase. Wild type and heterozygous *vmat2* larvae had more TH positive cells than homozygous *vmat2* larvae and **(B)** cell counts of wild type larvae (n = 12), heterozygous *vmat2 larvae (n = 8)* and homozygous *vmat2* larvae (n = 9) showed that this difference was statistically significant. The scale shows 100 μm.

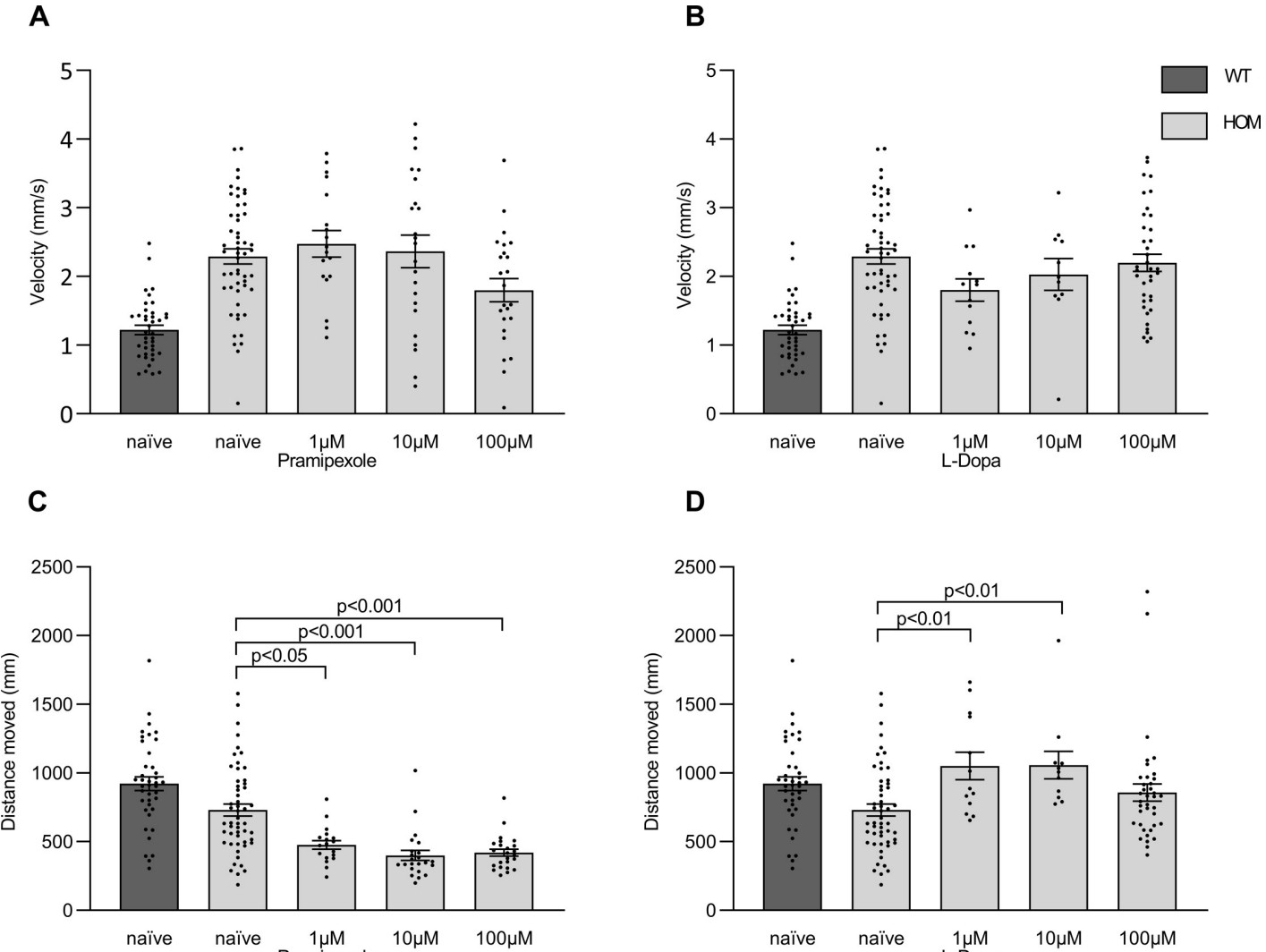

**Fig 3. Treatment of *vmat2* larvae with a dopamine agonist or precursor.** Homozygous *vmat2* (HOM) larvae were treated with three different concentrations (1μM, 10μM and 100μM) of pramipexole (**A, C**) or L-dopa (**B, D**) and compared to untreated larvae (naïve). Pramipexole and L-Dopa did not significantly alter the peak velocity at the onset of lights-off (**A, B**). Pramipexole (**C**) reduced the distance moved during lights-on significantly for all three concentrations whereas L-Dopa (**D**) increased it significantly for 1μM and 10μM. Naïve wild type (WT) larvae are included in the data for visual comparison. For values and statistics see Tables 1 and 2.

and homozygous mutant larvae in distance moved during lights-on, whereas L-Dopa rectified it (Fig 3C and 3D) (Table 2).

## Dopamine agonists and precursors have different effects on sleep across genotypes

We analysed the effects of the two anti-Parkinson´s compounds across multiple sleep parameters: sleep fragmentation, sleep ratio, velocity during sleep, average wake bout duration and average sleep bout duration, between the genotypes; both during the night and then during lights-on, following the night recording. First, during the night, in wild type larvae, pramipexole had no significant effects on sleep at any dose tested, whereas a low dose of L-Dopa tended to reduce sleep ratio and increase velocity across the night (Table 3). Similarly, in homozygous mutant larvae, pramipexole had minor effects on sleep (a single dose reduction in sleep

**Table 1. Peak velocity of wild type and homozygous *vmat2* mutant (HOM) larvae untreated (naïve) treated with pramipexole or L-Dopa.**

| Genotype | Drug | Dose | Peak velocity (mm/s) | | | |
|---|---|---|---|---|---|---|
| | | | Mean | ± | SEM | N |
| HOM | Naïve | | 2.29 | ± | 0.11 | 53 |
| | Pramipexole | 1μM | 2.47 | ± | 0.190 | 18 |
| | | 10μM | 2.37 | ± | 0.240 | 23 |
| | | 100μM | 1.80 | ± | 0.170 | 24 |
| | L-Dopa | 1μM | 1.80 | ± | 0.160 | 13 |
| | | 10μM | 2.03 | ± | 0.230 | 11 |
| | | 100μM | 2.20 | ± | 0.120 | 38 |
| WT | Naïve | | 1.22 | ± | 0.070 | 40 |
| | Pramipexole | 1μM | 1.23 | ± | 0.090 | 37 |
| | | 10μM | 1.07 | ± | 0.100 | 24 |
| | | 100μM | 1.11 | ± | 0.100 | 30 |
| | L-Dopa | 1μM | 1.22 | ± | 0.120 | 19 |
| | | 10μM | 1.53 | ± | 0.150 | 14 |
| | | 100μM | 1.13 | ± | 0.080 | 28 |

velocity) (Fig 4A), while L-Dopa significantly altered the sleep dynamics at multiple parameters. In homozygous mutant larvae, L-Dopa, changed sleep fragmentation in a dose-dependent manner, reduced sleep ratio, increased sleep velocity, lengthened average wake-bout durations and changed concentration dependent average sleep-bout durations (Fig 4B). Second, following lights-on, in wild type larvae, pramipexole increased sleep fragmentation, but massively reduced velocity while shortening wake bouts. On the other hand, L-dopa reduced fragmentation and sleep ratio, but increased velocity and wake bout duration (Table 3). In homozygous mutant larvae, pramipexole reduced velocity at all doses tested but only affected other parameters at mid dose (10μM) where it resulted in increased sleep ratio and sleep bout length and reduction in wake bout length (Fig 4C). L-dopa modulated fragmentation in an inverted-U shape curve, reduced sleep ratio and increased wake bout duration while changing sleep bout

**Table 2. Distance moved of wild type and homozygous *vmat2* mutant (HOM) larvae during lights-on.** Larvae were either left untreated (naïve) or treated with premipexole or L-Dopa.

| Genotype | Drug | Dose | Distance Moved (mm) | | | |
|---|---|---|---|---|---|---|
| | | | Mean | ± | SEM | N |
| HOM | Naïve | | 731 | ± | 43.7 | 53 |
| | Pramipexole | 1μM | 477 | ± | 31.7** | 18 |
| | | 10μM | 400 | ± | 36.9*** | 23 |
| | | 100μM | 420 | ± | 25.8*** | 24 |
| | L-Dopa | 1μM | 1051 | ± | 100.2* | 13 |
| | | 10μM | 1057 | ± | 100.7* | 11 |
| | | 100μM | 857 | ± | 62.5 | 38 |
| WT | Naïve | | 922 | ± | 50.0 | 40 |
| | Pramipexole | 1μM | 649 | ± | 27.2*** | 37 |
| | | 10μM | 427 | ± | 23.8*** | 24 |
| | | 100μM | 425 | ± | 64.3*** | 30 |
| | L-Dopa | 1μM | 1310 | ± | 77.5*** | 19 |
| | | 10μM | 1235 | ± | 95.6** | 14 |
| | | 100μM | 1106 | ± | 57.9 | 28 |

**Table 3. Sleep parameters of wild type and homozygous *vmat2* mutant larvae (HOM) untreated (naïve) or treated with premipaxole or L-Dopa during the night phase and following lights-on.**

| Recording period | Genotype | Drug | Dose | Sleep fragmentation (1/hr) | | | Sleep ratio | | | Sleep velocity (mm/s) | | | Wake duration (s) | | | Sleep duration (s) | | | |
|---|---|---|---|---|---|---|---|---|---|---|---|---|---|---|---|---|---|---|---|
| | | | | Mean | ± | SEM | Mean | ± | SEM | Mean | ± | SEM | Mean | ± | SEM | Mean | ± | SEM | N |
| **Night time** | **HOM** | **Naïve** | | 120.1 | ± | 3.09 | 0.59 | ± | 0.016 | 0.21 | ± | 0.005 | 12.2 | ± | 0.26 | 19.0 | ± | 1.050 | 53 |
| | | **Pramipexole** | 1µM | 123.0 | ± | 5.71 | 0.58 | ± | 0.035 | 0.20 | ± | 0.009 | 12.0 | ± | 0.57 | 18.0 | ± | 1.760 | 12 |
| | | | 10µM | 116.5 | ± | 6.66 | 0.64 | ± | 0.035 | 0.18 | ± | 0,008* | 10.9 | ± | 0.46 | 21.5 | ± | 2.390 | 14 |
| | | | 100µM | 130.6 | ± | 7.79 | 0.56 | ± | 0.048 | 0.19 | ± | 0.012 | 11.9 | ± | 0.63 | 16.6 | ± | 2.380 | 10 |
| | | **L-Dopa** | 1µM | 143.4 | ± | 4,39** | 0.38 | ± | 0,026*** | 0.26 | ± | 0,012*** | 15.6 | ± | 0,79*** | 9.8 | ± | 0,83** | 13 |
| | | | 10µM | 132.6 | ± | 4.89 | 0.49 | ± | 0,027* | 0.23 | ± | 0.012 | 14.1 | ± | 0,83* | 13.5 | ± | 1.040 | 11 |
| | | | 100µM | 101.7 | ± | 3,94*** | 0.64 | ± | 0.022 | 0.19 | ± | 0.006 | 12.4 | ± | 0.37 | 25.3 | ± | 1,89** | 38 |
| | **WT** | **Naïve** | | 127.3 | ± | 3.94 | 0.55 | ± | 0.016 | 0.21 | ± | 0.006 | 12.8 | ± | 0.26 | 16.8 | ± | 1.180 | 40 |
| | | **Pramipexole** | 1µM | 123.8 | ± | 3.07 | 0.54 | ± | 0.026 | 0.19 | ± | 0.007 | 13.2 | ± | 0.60 | 16.3 | ± | 1.090 | 26 |
| | | | 10µM | 127.2 | ± | 4.56 | 0.53 | ± | 0.038 | 0.19 | ± | 0.012 | 13.1 | ± | 0.68 | 15.7 | ± | 1.660 | 15 |
| | | | 100µM | 125.8 | ± | 5.87 | 0.52 | ± | 0.044 | 0.20 | ± | 0.011 | 13.4 | ± | 0.77 | 16.7 | ± | 2.350 | 20 |
| | | **L-Dopa** | 1µM | 125.6 | ± | 4.06 | 0.49 | ± | 0.019 | 0.24 | ± | 0,009** | 14.7 | ± | 0,47** | 14.5 | ± | 1.070 | 19 |
| | | | 10µM | 134.1 | ± | 5.96 | 0.49 | ± | 0.015 | 0.24 | ± | 0,007* | 13.9 | ± | 0.52 | 13.7 | ± | 1.080 | 14 |
| | | | 100µM | 112.3 | ± | 4,40* | 0.58 | ± | 0.02 | 0.20 | ± | 0.006 | 13.5 | ± | 0.35 | 20.1 | ± | 1.590 | 28 |
| **Day time** | **HOM** | **Naïve** | | 131.5 | ± | 3.05 | 0.41 | ± | 0.017 | 0.39 | ± | 0.018 | 17.1 | ± | 1.10 | 11.4 | ± | 0.520 | 53 |
| | | **Pramipexole** | 1µM | 141.2 | ± | 6.54 | 0.48 | ± | 0.054 | 0.25 | ± | 0,021*** | 13.1 | ± | 1.22 | 13.1 | ± | 2.150 | 12 |
| | | | 10µM | 138.6 | ± | 7.51 | 0.54 | ± | 0,045** | 0.22 | ± | 0,013*** | 11.5 | ± | 0,63* | 15.4 | ± | 1,89* | 14 |
| | | | 100µM | 143.6 | ± | 6.13 | 0.49 | ± | 0.049 | 0.22 | ± | 0,013*** | 12.9 | ± | 1.20 | 12.6 | ± | 1.750 | 10 |
| | | **L-Dopa** | 1µM | 106.9 | ± | 10,52** | 0.20 | ± | 0,032*** | 0.52 | ± | 0,051** | 35.4 | ± | 7,77*** | 6.2 | ± | 0,57** | 13 |
| | | | 10µM | 124.0 | ± | 4.95 | 0.30 | ± | 0,026* | 0.38 | ± | 0.018 | 20.8 | ± | 1.23 | 8.6 | ± | 0.820 | 11 |
| | | | 100µM | 118.4 | ± | 3,03* | 0.48 | ± | 0,022* | 0.29 | ± | 0,014*** | 16.0 | ± | 0.65 | 15.1 | ± | 0,94*** | 38 |
| | **WT** | **Naïve** | | 118.9 | ± | 4.64 | 0.33 | ± | 0.025 | 0.39 | ± | 0.018 | 22.3 | ± | 1.82 | 10.4 | ± | 0.950 | 40 |
| | | **Pramipexole** | 1µM | 134.4 | ± | 4,67* | 0.42 | ± | 0.034 | 0.25 | ± | 0,014*** | 15.5 | ± | 0,95** | 12.1 | ± | 1.320 | 26 |
| | | | 10µM | 143.0 | ± | 5,49** | 0.43 | ± | 0.041 | 0.22 | ± | 0,016*** | 14.2 | ± | 0,89** | 11.5 | ± | 1.450 | 15 |
| | | | 100µM | 140.3 | ± | 3,43** | 0.40 | ± | 0.039 | 0.25 | ± | 0,026*** | 15.4 | ± | 1,01** | 10.5 | ± | 1.200 | 20 |
| | | **L-Dopa** | 1µM | 97.2 | ± | 5,97* | 0.20 | ± | 0,033** | 0.51 | ± | 0,033** | 35.0 | ± | 5,55* | 7.2 | ± | 0.980 | 19 |
| | | | 10µM | 103.2 | ± | 7.53 | 0.27 | ± | 0.045 | 0.43 | ± | 0.036 | 30.3 | ± | 5.64 | 8.7 | ± | 1.130 | 14 |
| | | | 100µM | 121.1 | ± | 5.42 | 0.36 | ± | 0.024 | 0.34 | ± | 0.022 | 20.3 | ± | 1.56 | 11.5 | ± | 1.060 | 28 |

duration concentration dependent. At mid dose (10µM) L-dopa largely normalized the sleep parameters of homozygous mutant larvae to wild type levels (Table 3).

## GDNF brain injections rescue the aberrant peak velocity phenotype of *Vmat2* mutants

Due to the putative neuro-protective or -restorative effects of GDNF [29,42] we examined motor parameters such as peak velocity and distance moved following direct brain injection of GDNF. Injections of GDNF rescued the phenotype of increased peak velocity between wild type and homozygous mutant larvae. This was apparent when comparing GDNF injected larvae, sham injected larvae and naïve larvae, where a two-way ANOVA showed a statistical interaction between treatment and genotype ($F(2,178) = 5.707$, $p = 0.004$). A main effect was observed for treatment ($F(2, 178) = 10.41$, $p < 0.001$). This was followed up using Tukey´s multiple comparison post-hoc test which revealed differences between homozygous mutant and wild types for the naïve group ($1.74 \pm 0.23$ vs $1.04 \pm 0.10$, $p = 0.025$) and the sham group

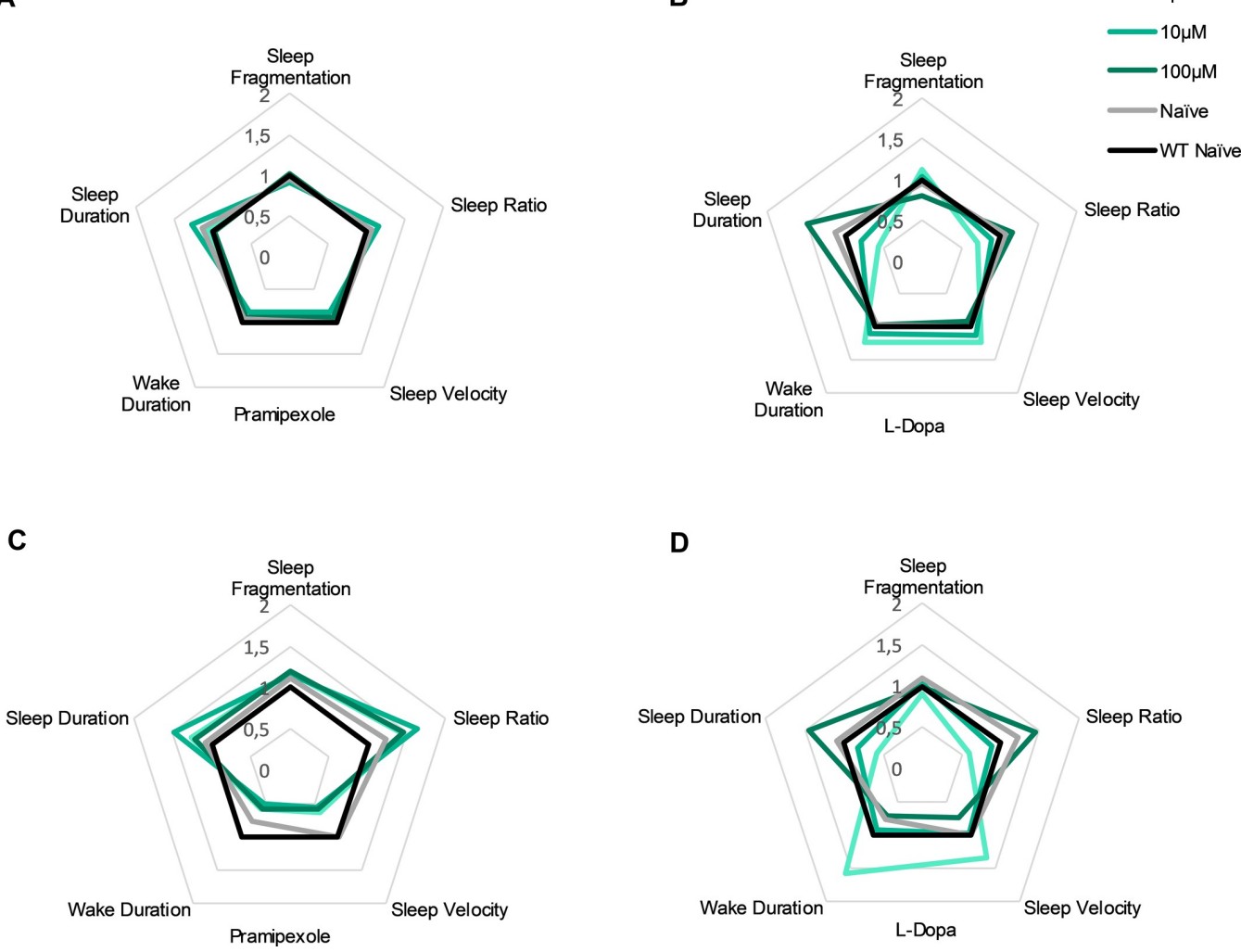

**Fig 4. Radar chart for sleep parameters of dopamine agonist or precursor treated *vmat2* larvae at night and day.** Homozygous *vmat2* (HOM) larvae were either left untreated (naïve, grey colour) or treated with three different concentrations (1μM, light green colour; 10μM, green colour; and 100μM, dark green colour) of pramipexole or L-Dopa and sleep parameters were analysed. Pramipexole did not significantly alter any sleep parameters while L-Dopa significantly altered the sleep dynamics at multiple parameters in dose dependent manner during the night **(A-B)** whereas during the day both compounds had immense dose-depended effects on all parameters **(C-D)**. Data are represented as fold change of untreated wild type (naïve, black colour) larvae. For values and statistics see Table 3.

$(2.12 \pm 0.21$ vs $1.13 \pm 0.13$, $p<0.001)$, but not for the GDNF group $(0.75 \pm 0.18$ vs $0.92 \pm 0.14$, $p = 0.987)$ (Fig 5). GDNF injections had no effects on distance moved (N.S.). These data suggest that GDNF can reverse the increase in initial movement during lights-off seen between the genotypes.

## Discussion

In the current study we characterize the behaviour of *vmat2* mutant zebrafish that display a salient, multifaceted phenotype, comparable to what has been described in *Vmat2* heterozygous mice [15,16] and human PD [15,16]. A photomotor assay revealed that all genotypes responded vigorously to lights-off transition; however, homozygous mutant *vmat2* larvae exhibited significantly higher peak-velocity following the transition, whereas the heterozygous

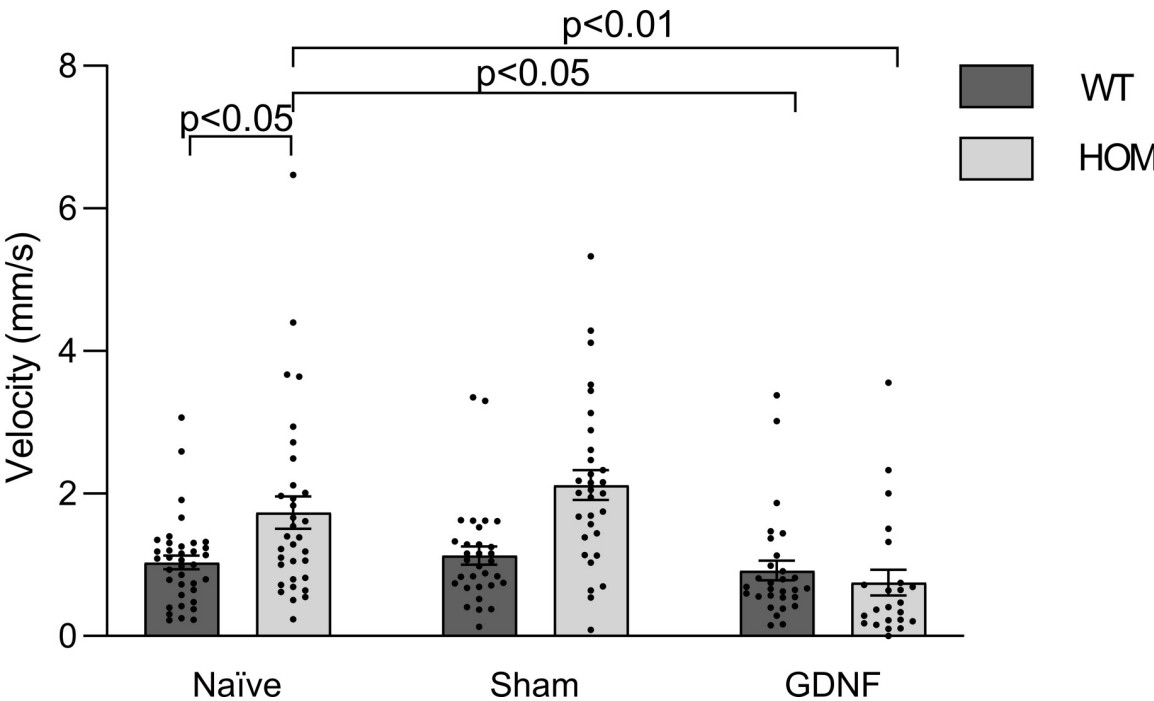

**Fig 5. Changes in peak velocity due to GDNF brain injections.** Wild type (WT) and homozygous *vmat2* (HOM) larvae were left untreated (naïve), injected with distilled water (sham) or injected with 1 ng/nl GDNF and peak velocity was examined at the onset of the first lights-off phase. GDNF injections rescued the phenotype of increased peak velocity between wild type and homozygous *vmat2* larvae: Wild type naïve group n = 37, homozygous *vmat2* naïve group n = 33, wild type sham group n = 31, homozygous *vmat2* sham group n = 31, wild type GDNF group n = 29 and homozygous *vmat2* GDNF group n = 23.

and wild type did not differ. Conversely, following lights-on transition the homozygous *vmat2* mutant larvae were hypomotile compared to either heterozygotes or wild type, which again, did not differ in their activity levels. The change in velocity was robust and consistent, and supports the data described recently for this mutant [27]. While the comparison between parkinsonian gait and balance and posture in zebrafish is challenging, important similarities remain. As human PD progresses towards advanced stages, episodic gait disturbances are observed as festination, start hesitation and freezing [43]. The homozygous *vmat2* mutants initiated movement with the strong and abrupt lights-off stimulus, but failed to inhibit movement to the same extent as wild type and heterozygous larvae. Conversely, during the lights-on period the homozygous mutant larvae were hypomotile.

Next, we assessed the differences between genotypes on sleep-wake cycles during the night. In human PD patients, sleep impairments are often reported as increased sleep fragmentation, excessive daytime sleepiness and REM behaviour disorder [44]. By comparing sleep ratio, average sleep and wake bout length, sleep fragmentation and velocity during the night, we revealed that the genotypes did not differ on any parameter during the night. In contrast, excessive day-time sleep ratio was evident in the homozygotes. Zebrafish express REM to non-REM alterations [45], however the behavioural sleep definition used to characterize sleep in zebrafish is limited to durations of sleep and wakefulness. It is therefore possible that differences in night-time sleep are missed, however a common sleep disorder concomitant with PD [46], excessive day-time-sleepiness, is present in the homozygotes. At mid dose (10μM) L-dopa largely normalized the sleep parameters of homozygous mutant larvae to wild type levels but with variable effects of all parameters depending on dose size. In wild types, pramipexole

administration induced higher sleep ratio as a result of more, but shorter, sleep bouts, whereas L-dopa induced lower sleep ratio as a result of fewer, but longer, wake bouts.

The *vmat2* mutant larvae exhibit an anxiety-like phenotype, revealed by avoidance of the arena centre and a preference for the close proximity to walls in a novel environment. Thigmotaxis has previously been validated in zebrafish larvae by exposure to anxiogenic substances [36]. An anxiety-like phenotype has been shown due to knockdown on vmat2 characterized by longer latency to the top in the novel tank test and dark-avoidance in the dark/light box test [26]. Similarly, increased thigmotaxis has been observed in mice following 6-OHDA chemical model of PD [47]. Thus, in the present study, homozygous *vmat2* mutants appear to exhibit enhanced anxiety-like behaviour, a hallmark feature of the non-motor symptoms of PD [48].

In a recent paper [34] we phenotyped in detail the established MPP+ chemical exposure model of PD in zebrafish. We showed that MPP+ incubated larvae swam less than control incubated larvae, but exhibited longer sleep bouts, more overall sleep and reduced thigmotaxis relative to them. They thus recapitulated aspects of the motor but not of the non-motor symptoms of human PD. In contrast, *vmat2* mutant larvae exhibit motor as well and non-motor deficits. The two different routes to disturb the neural circuits involved with PD thus generate non-identical phenotypes that may ultimately yield insights into the aetiology of the disease. Multiple genetic models of PD have been generated in zebrafish and have largely been focused on knock-down or knock-out of genes with known links with PD in humans and attempts to elucidate the neural effects that follow. These attempts include inactivation of synuclein, *Lrrk2*, *Parkin*, *Pink1* and *DJ1* [49]. These models have demonstrated reduction in dopamine cell count and transmission, impaired mitochondrial activity, hypersensitivity to toxins (including MPP+) and hypomotility [50–54]. The *vmat2* mutant shares some of the features (dopamine cell count and changes in motility) but their neural causes require fuller examination.

Previous studies on mice expressing very low levels of Vmat2 have shown greatly reduced dopamine levels in the brain [55,56], which led to the loss of TH-positive neurons [55]. Similarly, the recent study by [27] showed low expression of biogenic amines in *vmat2* mutant larvae (*vmat2*[-/-]) compared to sibling controls (*vmat2*[+/+]) [27]. The animal model investigated here therefore is postulated to have a non-functional Vmat2 protein, resulting in reduction in vesicular sequestration of monoamines and subsequent accumulation of toxic cytosolic monoamines. This subsequently will lead to a loss of dopamine cells. Indeed, we report a marked reduction in DA cell count in *vmat2* homozygous mutants.

In *vmat2* mutants we assume that DA receptors are functional and the catecholamine transporter is not. Therefore, we hypothesize that if the phenotype exhibited by *vmat2* mutants results primarily from diminished dopaminergic signalling, it would be rescued by the DA agonist pramipexole and that it would not be rescued and possibly worsened by L-Dopa; L-dopa treatment was found to worsen the symptoms in individuals homozygous for a hypomorphic allele of *VMAT2* [8]. However, our results did not match these predictions. Pramipexole had no significant effects on the peak velocity following lights-off (dark-light assay) and further decreased motility in the already hypomotile *vmat2* mutants. L-Dopa similarly had no effect on peak-velocity but partially rescued the hypomotility during lights-on. Turning to sleep, the trend continued with even greater differences: in wild type larvae neither pramipexole nor L-Dopa affected sleep parameters, whereas in homozygous *vmat2* larvae pramipexole had mild effects and L-Dopa greatly affected them. It is perhaps relevant that GABA can function as a co-transmitter in monoaminergic neurons and its release is dependent on Vmat2; thus some of the phenotypes in *vmat2* mutants may reflect altered GABAergic signalling and may interact with the PD drugs in unexpected ways [57]. A more reasonable interpretation is the effect is driven by the reduction of the other biogenic amines, also transported by Vmat2, instead of dopamine or interaction thereof. Finally, it is possible that genetic compensation,

perhaps induced by non-sense-mediated decay of the *slc18a2* transcript, influences the expression of homologous genes which impact the behaviour of these mutant larvae [58]. Jointly, the motor and sleep assays reveal a multifaceted genotype difference in the efficacy of the DA agonist and DA precursor.

Next, we examined the neurorestorative potential of glial cell-derived neurotrophic factor (GDNF) injected into the brain at 4 dpf. Neurotrophic therapy using GDNF has been investigated as a treatment for PD by replenishing neurotrophic factors, and shown to exert both neuroprotective and neurorestorative effects [29] in rhesus monkeys [59], rats [60] and rodents [61]. Unfortunately, in recent clinical trials GDNF treatment did not reach primary endpoints [62]. Thus, further research is required to establish the response to GNDF treatment. The zebrafish *vmat2* mutant may be useful in this research as GDNF injections, surprisingly, rescued the overreaction of the *vmat2* mutant larvae to lights-off stimuli but had no effects on other parameters. It is tempting to speculate that the rescue was due to a neuroprotective or restorative effect of the GDNF and further studies could reveal the molecular cascade leading to this rather specific effect.

In the current paper we describe a novel zebrafish *vmat2* mutant strain that is useful for studying this important carrier, may serve as a model for studying *VMAT2* function and its role in aetiology of PD. The homozygous *vmat2* mutant larvae exhibit a robust motor phenotype, increased thigmotaxis, abnormal day-time sleep behaviour and reduced DA cell count. The motor and sleep phenotype can only to a small extent be rescued in part by pramipexole and in part by L-Dopa whereas L-Dopa has immense effects on sleep behaviour. This model represents a novel valuable tool to perform high throughput pharmaceutical screens for therapeutics that interact with the Vmat2 protein, or substitute for it, and increase monoamine transport.

## Supporting information

**S1 Dataset.**
(XLSX)

## Author Contributions

**Conceptualization:** Karl Ægir Karlsson.

**Data curation:** Hildur Sóley Sveinsdóttir, Christian Christensen, Pablo Botella Lucena, Elena Richert, Robert Cornell, Karl Ægir Karlsson.

**Formal analysis:** Hildur Sóley Sveinsdóttir, Christian Christensen, Haraldur Þorsteinsson, Elena Richert, Karl Ægir Karlsson.

**Funding acquisition:** Robert Cornell.

**Methodology:** Amanda Decker, Valerie Helene Maier.

**Project administration:** Haraldur Þorsteinsson, Karl Ægir Karlsson.

**Resources:** Karl Ægir Karlsson.

**Supervision:** Robert Cornell, Karl Ægir Karlsson.

**Validation:** Amanda Decker.

**Visualization:** Karl Ægir Karlsson.

**Writing – original draft:** Valerie Helene Maier, Karl Ægir Karlsson.

**Writing – review & editing:** Hildur Sóley Sveinsdóttir, Amanda Decker, Christian Christensen, Haraldur Þorsteinsson, Valerie Helene Maier, Robert Cornell, Karl Ægir Karlsson.

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
