## [Decision Letter · Decision Letter 0]

12 Dec 2021

PONE-D-21-33867Motility phenotype in a zebrafish vmat2 mutantPLOS ONE

Dear Dr. Karlsson,

Thank you for submitting your manuscript to PLOS ONE. After careful consideration, we feel that it has merit but does not fully meet PLOS ONE’s publication criteria as it currently stands. Therefore, we invite you to submit a revised version of the manuscript that addresses the points raised during the review process.I am sorry for the delay in the reviewing process. One reviewer's comment was late. As you see from the attached comments, both reviewers only ask for minor revisions. I will leave the additional neurotoxic experiment at your discretion.

We look forward to receiving your revised manuscript.

Kind regards,

Stephan C.F. Neuhauss, Ph.D.

Academic Editor

PLOS ONE

Journal Requirements:

“I have read the journal's policy and the authors of this manuscript have the following competing interests: KÆK and HÞ are co-founders and shareholders in 3Z”

Reviewers' comments:

Reviewer's Responses to Questions

**Comments to the Author**

1. Is the manuscript technically sound, and do the data support the conclusions?

Reviewer #1: Yes

Reviewer #2: Yes

2. Has the statistical analysis been performed appropriately and rigorously? 

Reviewer #1: Yes

Reviewer #2: Yes

3. Have the authors made all data underlying the findings in their manuscript fully available?

Reviewer #1: Yes

Reviewer #2: Yes

4. Is the manuscript presented in an intelligible fashion and written in standard English?

Reviewer #1: Yes

Reviewer #2: Yes

5. Review Comments to the Author

Reviewer #1: This manuscript describes behavioral studies done on zebrafish with a targeted mutation in the slc18a2 (vmat2) gene. The mutants have been the object of previous publications by other groups but the aspects covered in the current manuscript are novel. The work is properly carried out.

I am moderately impressed by the magnitude of the effects observed by the authors. This may relate to the differential effects of the DA agonist and of the precursor. This surprising/interesting result is, however, extensively discussed by the authors. However, it may provide some warning as to the suitability of this model for drug screening. I would be hesitant to embark on a large-scale screen using this model and the studied parameters. This being said, it affects in no way the suitability of the manuscript for the journal.

I only have minor specific comments:

Lines 265 to 284 (page 10) The order of paragraphs in text does not follow the order of panels in Figure 1. Personally, I don’t have a problem with this but thought I would mention it.

Figure 2 : I have seen much better IHCs for TH than those shown in Figure 2. Can the authors provide better photographs?

Typos etc..

Line 87 : add space between “line” and “carries”

Line 338: “reduced”

Reviewer #2: The authors describe in detail the motility phenotype of vesicular monoamine transporter 2 (vmat2) by performing photomotpr assay, sleep assay, thigmotaxis assay on the mutant zebrafish larvae. Subsequently the authors used dopamine precursor and dopamine agonist as well as GDNF to observe potential rescue effects on the behavioral phenotype of the mutant larvae.

1. The authors need to compare and contrast the behavioural phenotype of vmat2 mutant zebrafish model with other zebrafish PD genetic models.

2. Did the authors try using a neurotoxin like MPTP/MPP+ on the vmat2 mutant larvae and analyzed the phenotype. It will be interesting to see the phenotype upon treating vmat2 mutant with a neurotoxin.

6. PLOS authors have the option to publish the peer review history of their article (what does this mean?). If published, this will include your full peer review and any attached files.

Reviewer #1: No

Reviewer #2: No

---

## [Author Response · Author response to Decision Letter 0]

17 Dec 2021

Dear Editor. What follows is a complete list of the changes made in the manuscript. First, the checklist provided by PLoS and, second, the specific comments raised by the reviewers. 

1. Plos Style policy: The manuscript now adheres to PLoS style policy. 

2. Funding information: I have verified that the grant numbers as stated in the manuscript are correct: GM067841 (RAC) and AR062547 (RAC). And the funding statements in the submission format and manuscript now match. 

3. Competing interests statment: The correct statement is now in the manuscript. 

4. Data availability: All data used in the paper is now submitted as S1. 

5. Data not shown: The phrase „data not shown“ has been removed. Now we simply state „N.S.“ since 0.05 has been defined as the significance level elsewhere in the manuscript. The relevant data is of course stored with full data set (see comment 4 above).

6. Reference section: References 49-54 have been added to the reference section (see comment a) from Reviewer #2). 

7. Reviewer #1 comments: 

a. Order of paragraphs P10: The order of the paragraphs have been changed to match the order of panels in Figure 1. 

b. IHC pictures: We feel that the IHC pictures are of sufficient quality to convey the key message of reduced DA cell count. 

c. Line 87: typo: Fixed. 

d. Line 338: typo: Fixed

8. Reviewer #2 comments: 

a. Contrast to other PD models: We added a short paragraph on comparison to other genetic PD models, and the appropriate references, to the fourth paragraph of the discussion. A full comparison of all zebrafish PD models is beyond the scope of this paper as can be seen in reference #49. 

b. MPTP/MPP+ additional experiment: The reviewer raises an interesting point. In fact, a subset of the authors have performed this experiment, as a part of seperate study that we cannot add to this one. But, the reviewer was indeed onto something since the vmat2 mutants seem to be hypersensitive to MPP+. 

We feel that we have fully and completely addressed all questions, comments and suggestions made. We thank the reviewers for their contribution in improving our manuscript.

---

## [Editor Report · Decision Letter 1]

20 Dec 2021

Motility phenotype in a zebrafish *vmat2* mutant

PONE-D-21-33867R1

Dear Karl,

We’re pleased to inform you that your manuscript has been judged scientifically suitable for publication and will be formally accepted for publication once it meets all outstanding technical requirements.

Kind regards,

Stephan C.F. Neuhauss, Ph.D.

Academic Editor

PLOS ONE
---

## [Editor Report · Acceptance letter]

26 Dec 2021

PONE-D-21-33867R1 

Motility phenotype in a zebrafish *vmat2* mutant 

Dear Dr. Karlsson:

I'm pleased to inform you that your manuscript has been deemed suitable for publication in PLOS ONE. Congratulations! Your manuscript is now with our production department. 

Kind regards, 

on behalf of

Dr. Stephan C.F. Neuhauss 

Academic Editor

PLOS ONE